# Molecular Characterization and Heterologous Production of the Bacteriocin Peocin, a DNA Starvation/Stationary Phase Protection Protein, from *Paenibacillus ehimensis* NPUST1

**DOI:** 10.3390/molecules24132516

**Published:** 2019-07-09

**Authors:** Chung-Chih Tseng, Lini Murni, Tai-Wei Han, Diana Arfiati, Hui-Tsu Shih, Shao-Yang Hu

**Affiliations:** 1Department of Dentistry, Zuoying Branch of Kaohsiung Armed Forces General Hospital, Kaohsiung 81357, Taiwan; 2Department of Marine Biotechnology and Resources, National Sun Yat-sen University, Kaohsiung 80424, Taiwan; 3Department of Biological Science and Technology, National Pingtung University of Science and Technology, Pingtung 91201, Taiwan; 4Department of Fisheries and Marine Science, University of Brawijaya, Malang 65145, Indonesia; 5Department of Environmental Biology and Fisheries Science, National Taiwan Ocean University, Keelung 20224, Taiwan; 6Department of Clinical pharmacy, Zuoying Branch of Kaohsiung Armed Forces General Hospital, Kaohsiung 81357, Taiwan; 7Research Center for Animal Biologics, National Pingtung University of Science and Technology, Pingtung 912, Taiwan

**Keywords:** bacteriocin, DNA starvation/stationary phase protection protein, fed-batch culture, *Escherichia coli*, antimicrobial activity

## Abstract

The production of a bacteriocin-like substance with antimicrobial activity, named peocin, by the probiotic *Paenibacillus ehimensis* NPUST1 was previously reported by our laboratory. The present study aimed to identify peocin and increase the peocin yield by heterologous expression in *Escherichia coli* BL21(DE3). Peocin was identified as a DNA starvation/stationary phase protection protein, also called DNA-binding protein from starved cells (Dps), by gel overlay and LC-MS/MS analysis. For mass production of peocin, fed-batch cultivation of *E. coli* was performed using a pH-stat control system. Purification by simple nickel affinity chromatography and dialysis yielded 45.3 mg of purified peocin from a 20-mL fed-batch culture (49.3% recovery). The biological activity of the purified peocin was confirmed by determination of the MIC and MBC against diverse pathogens. Purified peocin exhibited antimicrobial activity against aquatic, food spoilage, clinical and antibiotic-resistant pathogens. In an in vivo challenge test, zebrafish treated with purified peocin exhibited significantly increased survival rates after *A. hydrophila* challenge. The present study is the first to show the antimicrobial activity of Dps and provides an efficient strategy for production of bioactive peocin, which will aid the development of peocin as a novel antimicrobial agent with potential applications in diverse industries.

## 1. Introduction

Since the first antibiotic penicillin was accidentally discovered by Alexander Fleming in 1928, investigations on the isolation of new antibiotics or chemical modification of antibiotics have rapidly developed over the past few decades. To date, more than a thousand different antibiotics or modified antibiotics have been widely used in clinical medicine, rearing of livestock and poultry, aquaculture and agriculture for protection of humans and animals from pathogenic infections. However, the indiscriminate use of antibiotics has led to problems, including the rapid spread of multidrug-resistant (MDR) or pandrug-resistant (PDR) pathogens, reduced effectiveness of antibiotic treatment, damage to microbial ecosystems in the environment, and the emergence of antibiotic-resistant organisms from the consumption of antibiotic residues in food [1,2,3]. Given these problems, the development of alternatives to antibiotics for disease control is urgently needed.

Bacteriocins are defined as ribosomally synthesized antimicrobial proteins or peptides (AMPs) that are produced by bacteria and exhibit antagonistic activity against other bacteria for establishment of predominance in environmental microbial communities [4]. Unlike most antibiotics that are secondary metabolites, bacteriocins are proteinaceous compounds that are sensitive to proteases and generally harmless to the human body and surrounding environment. Moreover, due to the predominant properties of these substances, including a wide spectrum of antagonistic activities against pathogens, low toxicity, heat stability, and immunomodulatory effects on host immunity and because bacteriocins can be produced by generally regarded as safe (GRAS) probiotics, such as lactic acid bacteria, bacteriocins have been suggested as candidates for replacement of antibiotics [5]. Recently, bacteriocins and bacteriocin-producing bacteria have received attention for their applications in food preservation, biomedical therapy, rearing of livestock and poultry, and aquaculture [6,7,8]. For instance, fresh cheese inoculated with nisin-producing *Lactococcus lactis* or treated with pure enterocin exhibited significantly reduced amounts of the food-borne pathogen *Listeria monocytogenes* [9,10]. *Lactobacillus plantarum* with pediocin-like activity isolated from cattle shows potential probiotic traits of bile and low-pH tolerance, suggesting that this species could be used to overcome the problem of antibiotic resistance in veterinary medicine [11]. Bacteriocins such as nisin, pediocin A, and divercin were used to prevent necrotic enteritis in poultry caused by *Clostridium perfringens* [12]. Nisin-producing *Lactococcus lactis* strain A5 isolated from broadhead catfish (*Clarias macrocephalus*) exhibited antagonistic activity against pathogens, suggesting the potential applications of this strain as a feed component in aquaculture [13]. A novel bacteriocin-like inhibitory substance (BLIS) produced by *Enterococcus faecium* strain DSH20 exhibits antimicrobial activity against vancomycin-resistant *Enterococcus* (VRE) strains, suggesting the potential application of this BLIS to as an alternative antibacterial agent for the treatment of infections caused by drug-resistant VRE strains [14]. In summary, these reports demonstrated the application potential of bacteriocins or bacteriocin-producing bacteria as antibacterial agents in diverse industries. In general, bacteriocin-producing bacteria rather than pure bacteriocins are directly supplemented in feed for enhancement of animal health in the rearing of livestock and poultry and in aquaculture due to cost considerations. However, pure bacteriocins are usually used for food preservation or therapeutic purposes. Therefore, the production and purification of bacteriocins are important for potential commercialization and biomedical applications.

Currently, most pure bacteriocins are obtained from bacteriocin-producing bacteria via costly and time-consuming purification processes, such as ammonium sulfate precipitation combined with diverse chromatographic approaches. However, the ultimate recovery yield of bacteriocin is usually low due to the complexity of the purification process or due to low expression levels in the bacteriocin-producing bacteria. For instance, 305 µg of pure bacteriocin TSU (2.19% recovery yield) was obtained from a 1 L culture of *Lactobacillus animalis* TSU4 by a three-step purification process including ammonium sulfate precipitation, cation exchange chromatography, and reversed-phase chromatography [15]; 0.8 μg of the bacteriocin paracasein was acquired from a 1 L culture of *Lactobacillus paracasei* SD1 via purification by ammonium sulfate precipitation combined with chloroform precipitation and gel filtration chromatography [16]. The extremely low recovery yield from traditional purification processes makes large-scale production and application of bacteriocins challenging. The use of *Escherichia coli* as a cell factory provides an excellent platform for the expression of heterologous prokaryotic proteins. The well-established tools for the *E. coli* expression system, such as a variety of expression plasmids, a vast number of engineered strains, and many cultivation strategies, are easily accessible, and thus, this species is the most commonly used species for bacteriocin expression [17]. Recently, a report showed that the production yield of BtCspB, a novel cold-shock-like bacteriocin from *Bacillus thuringiensis*, by heterologous expression in recombinant *E. coli* significantly increased 449 times, supporting the use of *E. coli* as an effective system for bacteriocin production [18]. In our previous study, we isolated the probiotic *Paenibacillus ehimensis* NPUST1 with bacteriocin-like activity from tilapia culture ponds and demonstrated that the bacteriocin-like substance peocin exhibited heat stability, pH tolerance, and antagonistic activity against diverse aquatic pathogens, food spoilage bacteria, clinical pathogens, plant pathogens, and multiple antibiotic-resistant pathogens [19]. The present study aimed to characterize the bacteriocin peocin and optimize heterologous expression of peocin in *E. coli*. Moreover, a large-scale production strategy involving fed-batch cultivation of *E. coli* using a pH-stat and a simple purification process was developed to obtain biological peocin. We expected that the production and purification of peocin from recombinant *E. coli* could provide a model for effective enhancement of the yield of bacteriocin and provide insights for future application.

## 2. Results

### 2.1. Characterization of the Bacteriocin-Like Substance from P. ehimensis NPUST1

In our previous study, we demonstrated the production of a bacteriocin-like substance by *P. ehimensis* NPUST1 with properties of heat stability and pH tolerance. In the present study, the bacteriocin-like substance was identified by gel overlay analysis. The results showed an obvious band with a size of approximately 15 kDa that corresponded to the site of the inhibition zone on the tested agar plate (Figure 1A). To identify the protein, the protein band was excised and analyzed by LC/MS/MS (Appendix A). The mass spectra showed that the protein shared 97.26% coverage with the DNA starvation/stationary phase protection protein (accession no. 02585094) (Figure 1B,C). These results suggest that the bacteriocin-like substance peocin from *P. ehimensis* NPUST1 is a DNA starvation/stationary phase protection protein.

### 2.2. Expression and Characterization of the Recombinant Peocin by Flask Cultivation

To increase the yield of peocin, a DNA fragment corresponding to the amino acid sequence of peocin was synthesized and cloned into pET-28b, resulting in the pET-peocin plasmid, which was used to transform *E. coli* BL21(DE3) for heterologous expression of peocin. The N-terminus of peocin was tagged with six histidines for purification by a nickel-charged affinity resin (Figure 2). As shown in Figure 3, the *E. coli* culture produced a distinct recombinant peocin protein with a molecular weight slightly higher than 15 kDa. The recombinant peocin content with induction at 37 °C was approximately 51.6% of the total protein content (Figure 3A lanes 1 and 2), and more than 82.6% of the total recombinant peocin was expressed in soluble form (lanes 3 and 4). Cultivation of the recombinant *E. coli* strain at a relatively low temperature (27 °C) was attempted to increase the level of soluble peocin; however, the effect was limited (lanes 5 and 6). The identity of the peocin protein expressed in recombinant *E. coli* after IPTG induction was further confirmed by Western blotting using a 6x-His epitope tag antibody. A significant signal that was absent in the noninduced sample confirmed the identity of the recombinant peocin (Figure 3B). To preliminarily test the antimicrobial activity of recombinant peocin, the crude protein-containing supernatant from recombinant *E. coli* cells carrying the pET-peocin plasmid was subjected to a disc diffusion test against *A. hydrophila*. The results showed a distinct inhibition zone, convincingly demonstrating that peocin is a bacteriocin-like substance produced by *P. ehimensis* NPUST1 and that the recombinant peocin possesses antimicrobial activity (Figure 3C).

### 2.3. Fermentation of Recombinant E. coli by Fed-Batch Cultivation

To scale-up the production of the peocin protein, pH-stat-based fed-batch cultivation was used for high-cell-density production of recombinant *E. coli* in a 5-L bioreactor. The initial glucose (20 g/L) in the bioreactor was gradually consumed, accompanied by cell growth during the batch culture period. Feeding with 28% ammonium hydroxide (*v/v*) was activated to maintain a natural pH and supply a nitrogen source during cultivation when the pH value was less than 6.9. After 7.5 h of cultivation, the initial glucose was exhausted, and the addition of feeding solution was triggered due to the increase in pH to a value higher than the set value of 7.1. Subsequently, the culture temperature was shifted to 30 °C from 37 °C to avoid accumulation of the byproduct acetate. Because glucose was the limiting parameter during fed-batch cultivation, the glucose concentration was maintained below 1.8 g/L, and the acetate concentration was maintained below 1.5 g/L during the complete cultivation process. A high biomass of 30.2 g/L DCW was obtained after 30 h of cultivation, and more than 92% plasmid stability was observed at the end of cultivation (Figure 4A). To evaluate the production efficiency of recombinant peocin at the time of induction, cells were induced with 0.1 mM at the early exponential phase (6 h), midexponential phase (12 h), and late exponential phase (18 h) of growth. When cells were induced in the early exponential phase, the peocin content increased rapidly for 9 h post induction, and then, a constant expression level of peocin was maintained until the end of cultivation. The DCW and recombinant peocin content at the end of cultivation were 15.3 ± 0.82 g/L and 32.1 ± 1.32%, respectively (Figure 4B). When cells were induced at midexponential phase, the peocin content increased for 6 h post induction and then decreased slightly until the end of cultivation. The DCW and recombinant peocin content at the end of cultivation were 23.2 ± 1.44 g/L and 14.7 ± 1.11%, respectively (Figure 4C). When cells were induced in the late exponential phase, the peocin content increased for 6 h post induction, and then, a stable expression level was maintained until the end of cultivation. The DCW and peocin content were 29.7 ± 1.57 g/L and 16.6 ± 1.21%, respectively (Figure 4D). Although cells induced during the late exponential phase had a higher DCW than those induced during the early and midexponential phases, the peocin content was low. Cells induced in the early exponential phase have the highest expression level of peocin and low biomass, which is beneficial for subsequent purification. Thus, induction of cells at the early exponential phase was selected as the optimal condition for the production of peocin, and the concentration of peocin reached as high as 3 g/L.

### 2.4. Purification and Characterization of Recombinant Peocin

Purification of recombinant peocin was performed from 20 mL of culture broth from a fed-batch culture induced at the early exponential phase using Ni-NTA affinity chromatography. The recovery yield from each purification step is provided in Table 1. The final amount of peocin and the recovery yield were 45.3 mg and 49.3%, respectively. The purity of purified peocin as determined by HPLC was 92.1% (Figure 5A,B). The spectrometrically determined mass of the purified peocin was 17,588.4 Da, which is consistent with the molecular mass of peocin plus the amino acids generated by the restriction site and the six histidines (Figure 5C).

### 2.5. Biological Activity of Purified Peocin

To determine whether the purified peocin exhibited biological activity, the in vitro antimicrobial activity of peocin against diverse pathogens was determined by MIC and MBC assays. As shown in Table 2, the purified peocin exhibited antagonistic effects against the aquatic pathogens *A. hydrophila*, *Streptococcus agalactiae*, *Streptococcus iniae*, *Vibrio vulnificus*, *Vibrio parahaemolyticus*, *Vibrio alginolyticus*, and *Debaryomyces hansenii*; food spoilage bacteria *Escherichia coli*, *Staphylococcus aureus*, *Salmonella typhimurium*, and *Listeria monocytogenes*; clinical pathogens methicillin-resistant *S. aureus* (MRSA), *Pseudomonas aeruginosa*, *Porphyromonas gingivalis*, *Prevotella nigrescens*, and *Propionibacterium acnes*. Although the bacteriostatic and bactericidal potency of peocin were lower than those of the antibiotic kanamycin, the antimicrobial activity of peocin against the antibiotic-resistant pathogens *V. parahaemolytics*, *V. alginolyticus*, and MRSA was impressive. The in vivo antimicrobial activity of peocin was also evaluated in a zebrafish model via *A. hydrophila* challenge. First, the optimal dose of peocin for treatment of zebrafish was evaluated by injecting 1 µg, 5 µg, 10 µg and 20 µg of peocin per fish. Compared with the group without peocin treatment, significant mortality of zebrafish was observed at doses of 10 and 20 µg of peocin per fish, and no significant difference was observed at doses lower than 5 µg of peocin per fish (Figure 6A). Thus, injected doses of 1 and 5 µg of peocin per fish were used to evaluate the protective efficacy of peocin in zebrafish after *A. hydrophila* infection. The zebrafish survival rate at 7 days post infection was 100% for the group injected with saline. However, the survival rates of *A. hydrophila*-infected zebrafish at 5 post infection days were dramatically reduced, with a value of 41.6 ± 5.77%. Notably, the survival rates at 7 days post infection were 63.3 ± 7.63% and 71.67 ± 2.88% for the group injected with 1 and 5 µg of peocin per fish, respectively, and *A. hydrophila*. The cumulative survival of fish treated with 1 and 5 µg of peocin per fish was significantly enhanced compared to that of fish that were not treated with peocin (Figure 6B).

## 3. Discussion

The emerging trend of the development of bacteriocins as alternatives to antibiotics or immunostimulators is accompanied by the rapid expansion of studies on isolation, genetics, engineering production, and potential applications of bacteriocins [4]. In our previous study, we reported that *P. ehimensis* NPUST1 exhibited bacteriocin-like activity and demonstrated the immunomodulatory and disease resistance effects of this organism on tilapia (*Oreochromis noltica*) [19]. In the present study, the bacteriocin-like substance peocin was identified as a DNA starvation/stationary phase protection protein by gel overlay and LC-MS/MS analysis. The DNA starvation/stationary phase protection protein, also called DNA-binding protein from starved cells (Dps), is widely distributed in prokaryotes, and the levels of this protein increase dramatically during the stationary phase. The archetypical function of Dps is to form Dps-DNA complexes and protect DNA from hydroxyl radicals generated by the intracellular reaction of free iron with H_2_O_2_ [20,21]. In addition to DNA binding activity, Dps also play important roles in protecting cells against diverse forms of stress, such as starvation, oxidative stress, metal toxicity, and thermal stress [22]. However, the antimicrobial function of Dps has never been reported previously; thus, the production of pure peocin to investigate the biological function and potential application of this protein is necessary. The present study showed that peocin was expressed in engineered *E. coli* and attempted to demonstrate the antibacterial function of this protein. As shown in Figure 3, the supernatant from recombinant *E. coli* harboring the pET-peocin expression plasmid exhibited a distinct inhibition zone against *A. hydrophila* that did not exist for the supernatant from *E. coli* harboring the pET-peocin without IPTG induction, demonstrating that the bacteriocin peocin possesses antibacterial activity. To the best of our knowledge, this is the first report to show that Dps possesses antimicrobial activity. Chuancone and Ceci have reported that Dps family members contain a ferroxidase center that is highly conserved among more than 300 bacterial species [21]. However, whether the antibacterial function and mechanism of action of Dps are common properties in bacteria remains unclear and is worth investigating in the future.

The production of various recombinant proteins, such as hormones, enzymes, antimicrobial proteins, and protein drugs, by the *E. coli* expression system has been widely applied in the biotechnology and pharmaceutical industries. In the present study, the plasmid pET-peocin was transferred into *E. coli* BL21(DE3) for expression of recombinant peocin, and most of the recombinant peocin was expressed in soluble form in *E. coli*. Although reports have shown that slowing of bacterial growth to retard protein synthesis and folding by reduction of temperature is an efficient approach to increase the soluble proportion of recombinant proteins [23,24], this strategy is invalid in this case. For mass production of peocin from recombinant *E. coli*, an alternative to flask cultivation is required. Fed-batch cultivation of *E. coli* via a pH-stat-based strategy provided an easy and efficient approach for manipulation to improve the biomass production yield [25]. The feeding of nutrients was activated based on an increase in pH beyond the set threshold due to the depletion of glucose. This feeding mechanism ensures that the fed-batch culture can maintain a low glucose concentration (below 1.8 g/L) to prevent the accumulation of acetic acid, which is overproduced under oxygen-limiting conditions and is well known as an inhibitor of cell growth and recombinant protein expression in *E. coli*. Several studies have reported high-cell-density cultivation (>100 g/L DCW) of recombinant *E. coli* by supplying pure oxygen to prevent acetate accumulation [25,26]; however, the high cost and safety risks associated with pure oxygen lead to challenges in the scale-up of this process for industrial applications. Herein, an efficient fermentation process without supplying pure oxygen was developed for the mass production of recombinant *E. coli*. A 20% air saturation was maintained by decreasing the cultivation temperature during fed-batch cultivation and increasing the agitation speed. The acetic acid concentration during the entire cultivation period could be sustained effectively at less than 1.5 g/L, and a cell density of 30.2 g/L DCW was obtained after 30 h of cultivation under the optimal culture conditions. The efficiency of peocin expression was estimated by IPTG induction at different growth phases. Regardless of induction time, the cell mass was significantly lower than that without IPTG induction due to the metabolic burden of peocin expression. Thus, the lowest cell mass and highest peocin expression level were obtained when induction was carried out at the early exponential phase. Although a relatively high biomass was obtained by induction at the late exponential phase, cells at high density may exhibit response to various forms of stress, such as nutritional deficiency, accumulation of metabolic waste, and osmotic pressure imbalance, harming the expression of recombinant proteins [26]. Therefore, a relatively low peocin expression level was obtained by induction at the middle or late exponential phase. In the present study, the maximum amount of peocin produced was approximately 3 g/L (32.1% of total protein) when cells were induced at the early exponential phase. Although certain bacteriocins have been cloned and expressed in *E. coli*, the present study is the first to report mass production of bacteriocin by fed-batch cultivation without pure oxygen supplementation.

A simple and effective purification process is an important and critical factor for acquisition of enough recombinant protein and to achieve scale-up goals in industry. Previous reports have demonstrated that bacteriocins could be efficiently purified by affinity chromatography, suggesting the feasibility of affinity chromatography for peocin purification [18,27,28]. Moreover, fusion of the 6xHis tag to bacteriocins has been suggested to retain the native structure and biological activity of the protein [28]. In the present study, peocin purification was performed by a simple Ni-NTA affinity chromatography step. A total of 45.3 mg of bioactive peocin (recovery yield, 49.3%) with 92.1% purity was acquired from a 20-mL fed-batch culture broth, indicating that 2.26 g of purified peocin can be obtained from 1 L of fed-batch culture. The molecular weight (MW) of the purified peocin matched the predicted MW of recombinant peocin and confirmed the identity of the purified peocin. Bacteriocin have been classified into four classes based on their structure, molecular weight and properties, and post-translational modifications. Class I bacteriocins typically are extensively post-translationally modified peptides, which can be further subdivided into class Ia (lantibiotics), class Ib (labyrinthopeptins), and class Ic (sanctibiotics). Class II are small non-modified peptides that can be further sub-divided into class IIa (pediocin-like bacteriocins), class IIb (two-peptides unmodified bacteriocins), class IIc (circular bacteriocins), and class IId (unmodified, linear, non-pediocin-like bacteriocins). Class III bacteriocins consists of large and heat-labile proteins for which there is only scarce information available. Class IV bacteriocins consists of large complexes of protein with carbohydrate or lipid were re-named as bacreiolysins and removed from the bacteriocin category [29]. Based on the criteria of classification, the heat-stable peocin bacteriocin with molecular weight of 17588.4 Da is considered as Class III bacteriocin. Diverse bacteriocins expressed by recombinant *E. coli* have been studied and reviewed [17]. In the present study, the highest bacteriocin yield was obtained with fed-batch cultures of *E. coli* BL21 (DE3)/pET-peocin in a 5 L controlled bioreactor (Table 3). The significant increase of peocin production compared to other bacteriocins production was potentially due to the increasing of cells production and higher expression level of peocin in *E. coli.*

In the present study, the biological activity of the purified peocin was confirmed by demonstrating the bacteriostatic and bactericidal activity of this protein against diverse pathogens and protective effects against bacterial infection in zebrafish. The purified peocin exhibited wide-spectrum antimicrobial activity against aquatic pathogens, food spoilage bacteria, clinical pathogens, and multiple antibiotic-resistant pathogens, suggesting a potential application of peocin in food preservation and as an alternative to antibiotics for disease control in humans and animals. Interestingly, we found that *E. coli* cells continued to grow after IPTG induction of peocin expression, as shown in Figure 4, indicating that peocin within cells did not cause mortality in *E. coli*. However, purified peocin revealed antimicrobial activity against *E. coli,* as shown in Table 2. The result indicates that the intracellular and extracellular roles and functions of Dps may differ; however, this hypothesis must be further studied and demonstrated. In the last decade, a variety of bacteriocin isolated from bacteria are reported with antimicrobial activity against diverse pathogens. Moreover, a few bacteriocins, i.e., nisin, pediocin PA-1/AcH, and lacticin 3147 are being used in the food industry [30]. However, the evaluation of toxicity are the crucial factors required for bacteriocins to be used in industry. Zebrafish has been commonly used as an animal model to evaluate the toxicity and efficacy of compounds [31,32]. The present study revealed that the injected dose of peocin higher than 5 µg/fish is toxic to fish, suggesting toxicity assessment of peocin is required to ensure its safety before practical application in industry. IP injection with 1 or 5 µg of peocin per fish before *A. hydrophila* infection significantly improved fish survival, suggesting that peocin exhibits potent inhibition of bacterial infection. The protective effects of peocin against pathogenic infection in fish provide information regarding the efficacy of peocin that will prove useful in possible therapeutic applications. In conclusion, this is the first study to elucidate the antimicrobial activity of Dps. An efficient procedure was established for the production and purification of the bacteriocin peocin. The effective antimicrobial potency against diverse pathogens, including antibiotic-resistant pathogens, suggests the potential application of antibiotics as novel antibacterial agents for disease control in animal rearing and biomedicine.

## 4. Materials and Methods

### 4.1. Fish and Bacterial Strain

Adult AB strain zebrafish (*Danio rerio*) with body lengths of 3.67 ± 0.3 cm and body weights of 0.72 ± 0.11 g were acquired from the Taiwan Zebrafish Core Facility at Academia Sinica (Taipei, Taiwan). The fish were acclimated in a 120-L freshwater recirculating tank (14 h light/10 h dark) at 28 °C in an aquatic laboratory animal facility at National Pingtung University of Science and Technology (NPUST), which is recognized by the Association for Assessment and Accreditation of Laboratory Animal Care (AAALAC) and fed daily with a commercial diet (MeM Prime, BERNAQUA, Olen, Belgium). The experiments were approved by the NPUST Institutional Animal Care and Use Committee (IACUC approval No. NPUST-106-070) and performed in compliance with local animal welfare regulations. Anaerobic pathogens *Porphyromonas gingivalis* (BCRC 14417), *Propionbacterium acnes* (BCRC 10723) and *Prevotella nigrescens* (BCRC 14408) were purchased from the Bioresource Collection and Research Center (BCRC) (Hsinchu, Taiwan), and the culture conditions and protocols followed the instruction manual. The source of the other pathogens and the culture methods used in this study are described in a previous report [19].

### 4.2. Gel Overlay and Mass Spectrometry Analysis

The protocol for cultivation of *P. ehimensis* NPUST1 was described previously in [19]. The bacteriocin-like substance from cell-free culture broth after 96 h of cultivation was partially purified by ammonium sulfate precipitation. Subsequently, the partially purified bacteriocin-like substance was separated by 15% SDS-PAGE, and the gel was then placed on a TSB agar plate with *Aeromonas hydrophila*. After cultivation at 28 °C for 16 h, the protein band on the SDS-PAGE gel corresponding to the inhibition zone on the TSB agar plate was excised for in-gel digestion. In-gel digestion was performed according to the protocol described by Chung et al. [39]. The dried peptide mixtures were reconstituted in HPLC buffer A (0.1% formic acid) and loaded onto a reversed-phase column (Zorbax 300SB-C18, 0.3 × 5 mm; Agilent Technologies, Wilmington, DE, USA). The desalted peptides were then separated on an in-house column (HydroRP 2.5 μm, 75 μm I.D. × 20 cm with a 15-μm tip) using a linear gradient of 6–29% HPLC buffer C (99.9% acetonitrile/0.1% formic acid) for 46 min, 29–36% buffer C for 7 min, 36–95% buffer C for 5 min, 95% buffer C for 3 min, and 2.5% buffer C for 4 min, at a flow rate of 0.3 μL/min. The LC apparatus was coupled to a 2D linear ion trap mass spectrometer (Orbitrap Elite; Thermo Fisher, San Jose, CA, USA) operated using Xcalibur 2.2 software (Thermo Fisher, San Jose, CA, USA).

### 4.3. Expression Plasmid and Flask Cultivation

The peocin gene that encodes amino acids corresponding to the DNA starvation/stationary phase protection protein (Dps) (accession no. 02585094) was synthesized by GenScript (Piscataway, NJ, USA) and ligated into the pET-28b vector to form a pET-peocin expression plasmid using the *Nco*I and *Xho*I restriction enzyme sites. Expression of the peocin gene was under the control of the T7 promoter and induced by isopropyl-β-D-thiogalactopyranoside (IPTG) (Sigma, St. Louis, MO, USA). *E. coli* DH5α was used as a host for maintenance of the expression plasmid. *E. coli* BL21(DE3) was used as a host for expression of the peocin protein. Growth of the stock and seed cultures of recombinant *E. coli* BL21(DE3) was performed according to a previously protocol described study [25]. Shake flask cultivation was conducted by inoculating 1 mL of seed culture into a 500-mL flask containing 100 mL of modified SSP medium (15 g/L peptone, 5 g/L yeast extract, 5 g/L glucose, 8 g/L K_2_HPO_4_, 2 g/L KH_2_PO_4_; pH 7.5) and incubated at 37 °C and 150 rpm. Following a 3-h cultivation (OD_600_ = 1), recombinant peocin expression was induced by adding IPTG at a final concentration of 0.1 mM. After 12 h of cultivation, an appropriate volume of culture broth was sampled and harvested by centrifugation at 4000× *g* for 15 min at 4 °C. The cell pellet was resuspended in a 15-mL centrifuge tube containing 2 mL of PBS, resulting in a cell suspension with an OD_600_ of 30. Cell lysis was performed by using a sonicator (Chrom Tech. UP-800, Taiwan) at 20% power for 90 sec. The peocin content in the supernatant and pellet was evaluated by 15% SDS-PAGE.

### 4.4. Bioreactor and Fed-Batch Cultivation

For high-level production of recombinant peocin, fed-batch cultivation with a pH-stat was conducted in a 5-L bioreactor (Winpact FS-02, Taiwan) equipped with a built-in digital controller for pH, temperature, agitation, dissolved oxygen (DO), and antifoam and peristaltic pumps for addition of acid, base, antifoam, and nutrients. The composition of the modified R medium and protocol for manipulation of the fed-batch culture were previously described in [39]. Briefly, in the 5-L bioreactor, 2.5 L of modified R medium was used as the initial working volume. The initial agitation speed and flow rate of aeration were set at 500 rpm and 3 L/min. The agitation speed was gradually increased to the upper limit of 900 rpm accompanied by reduction in DO to maintain 20% air saturation. The pH was maintained at 7.0 by adding 28% (*v/v*) ammonium hydroxide. After 2 h of agitation and aeration, the DO in the medium was set to 100% air saturation. A 100-mL seed culture (4% *v/v*) was inoculated into the initial working volume. Feeding with nutrient solution (750 g/L glucose, 50 g/L yeast extract, and 75 g/L peptone) was automatically activated to supplement nutrients in the culture broth, when the pH increased to more than 0.1 U beyond the set value (pH 7.0) due to depletion of glucose in the initial medium. When fed-batch feeding was activated by the initial glucose consumption, the cultivation temperature was shifted from 37 °C to 30 °C to avoid acetate accumulation resulting from the high growth rate. Expression of the peocin protein was induced with 0.1 mM IPTG. Sterilized antifoaming agent (Sigma A-5758, St. Louis, MO, USA) was automatically added to suppress excess foaming. Culture samples were withdrawn periodically for quantitative analysis during cultivation. Cell growth was monitored by measuring the optical density at 600 nm (OD_600_) on a spectrophotometer (Thermo, G10S UV–Vis, Madison, WI, USA). Dry cell weight (DCW), plasmid stability, and glucose and acetate concentrations were determined as reported by Chung et al. [39]. The expression levels of recombinant peocin were analyzed on a 15% SDS-PAGE gel stained with Coomassie brilliant blue R-250 (Bio-Rad). Peocin was quantified using a densitometer (BioSpectrum 500 Imaging System, UVP, CA, USA) to scan the peocin band in the gel. The protein concentration was determined by the Bradford protein assay method using BSA as the standard [40].

### 4.5. Purification of Recombinant Peocin Protein

A 20-mL aliquot of the culture broth from the fed-batch culture that was induced at the early exponential phase was collected by centrifugation at 4000× *g* for 15 min at 4 °C. The harvested cell pellet was washed two times in 40 mL of PBS (140 mM NaCl, 2.7 mM KCl, 10 mM Na_2_HPO_4_·12H_2_O, 1.8 mM KH_2_PO_4_; pH 7.3) and centrifuged again. The washed cell pellet was resuspended thoroughly in 25 mL of PBS and lysed using a sonicator (Chrom Tech. UP-800, Taiwan) at 20% power output. The lysed cell solution was centrifuged at 17,500× *g* for 30 min at 4 °C to collect the supernatant fraction and then loaded on a 5-mL Ni-NTA agarose resin column (Qiagen, Valencia, CA, USA) for specific binding of the recombinant peocin protein. The resin was washed with 50 mL of PBS containing 30 mM imidazole, and the protein was eluted with 15 mL of PBS containing 250 mM imidazole. The eluate containing purified peocin was placed in a 3.5 kDa cut-off dialysis bag and dialyzed against 500 mL of PBS (pH 7.3) at 4 °C for 16 h and then centrifuged at 17,500× *g* for 30 min at 4 °C to remove the aggregated protein. The purity of purified peocin was analyzed by reverse-phase HPLC equipped with a betabasic-C18 column (4.6 mm × 150 mm, 5 µm, Keystone Scientific Inc., Bellefonte, PA, USA). Proteins were separated using a linear gradient from 30% in buffer A (0.1% TFA in H_2_O) to 100% in buffer B (0.1% TFA in acetonitrile) at a flow rate of 0.2 mL/min, and detection was achieved by monitoring the UV absorbance at 220 nm. The purified peocin was stored at 4 °C and prepared for antibacterial activity analysis.

### 4.6. Molecular Mass Determination

The supernatant containing peocin obtained by affinity chromatography was further purifietd by reversed-phase HPLC (in-house column, 200 mm × 150 μm, Jupiter 5 μm C4 300 Å, Phenomenex, Torrance, CA, USA). Charge deconvolution was performed using ProMass (Thermo, Waltham, MA). The average spectrum (896–905) was imported to ProMass to calculate the mass of the purified peocin protein.

### 4.7. MIC and MBC Determination

The MIC and MBC values of the purified peocin were determined using a modified resazurin microplate assay according to the protocol described in [41]. In brief, 100 µL of reaction mixtures containing 10 µL of the indicator pathogen (10^7^ CFU/mL), 67 µL of TSB medium, 13 µL of resazurin solution (0.25 mg/mL) and 10 µL of various concentrations of purified peocin were placed in 96-well microplates and cultured at 28 °C for 12 h. Serial dilutions of kanamycin (Sigma-Aldrich, K-4000, St. Louis, MO, USA) with the corresponding bacterial strains were used as positive controls. The MIC value was defined as the lowest peocin concentration that prevented a color change in the test well. The MBC was determined by taking the entire sample from each well with a dark green color and streaking it on a TSB agar plate and then incubating the sample at 28 °C for 24 h. The MBC value was defined as the lowest peocin concentration at which colonies did not form on the agar plate.

### 4.8. Challenge Test

The protocol of the challenge test was modified from a previous report [42]. Briefly, overnight *A. hydrophila* cultures were centrifuged to collect bacterial cells. The cell pellets were suspended in sterile water and diluted to a concentration of 1 × 10^7^ CFU/mL. The bacterial challenge experiment was conducted in triplicate by intraperitoneal (IP) injection of 10 µL of diluted *A. hydrophila* (1 × 10^5^ CFU per fish, equivalent to 7 days of the fifty percent lethal dose (LD_50_)): into 20 zebrafish. The effect of purified peocin on the cumulative survival of the pathogen-infected fish was evaluated by injecting each fish with 10 µL of various concentrations of purified peocin or 10 µL of purified peocin (1 or 5 µg/fish) 2 h prior to injecting 10 µL of the pathogen. Experimental fish (20 fish per tank) were kept in tanks containing 20 L of freshwater at 28 °C. Fish injected with pathogens or sterile water were used as positive or negative controls. The challenged fish were observed daily, and the cumulative survival in each tank was recorded for 7 days. The cumulative survival data were analyzed by the Kaplan–Meier method using SAS software (SAS Institute, Cary, NC, USA). 

## Figures and Tables

**Figure 1 molecules-24-02516-f001:**
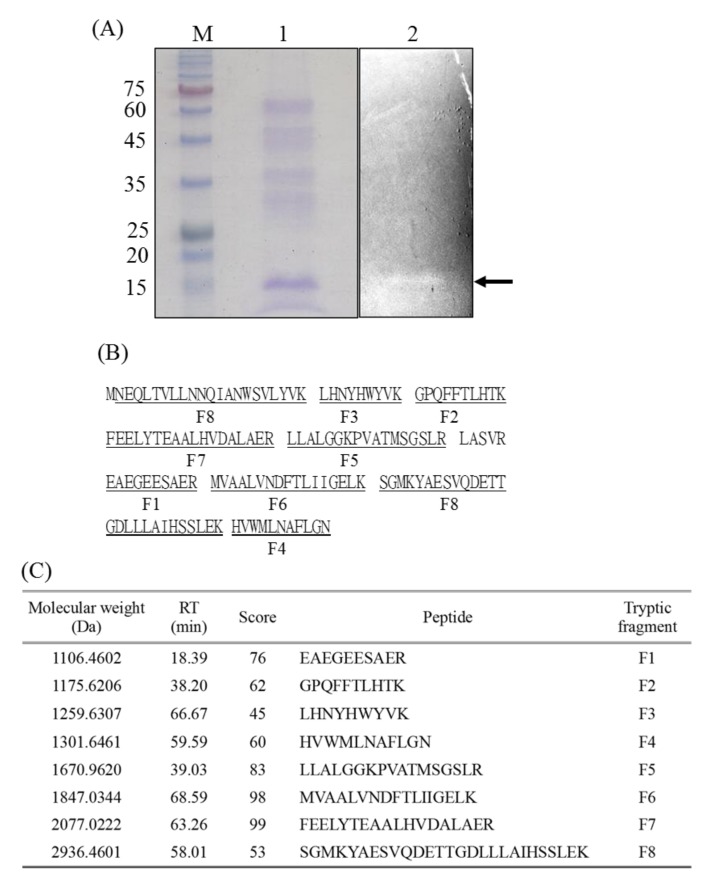
Characterization of the bacteriocin-like substance from *Paenibacillus ehimensis* NPUST1. (**A**) Gel overlay analysis. Lane M: marker; lane 1: SDS-PAGE analysis of ammonium sulfate-precipitated cell-free culture broth of *P. ehimensis* NPUST1; lane 2: SDS-PAGE gel placed on a TSB agar surface overlaid with *Aeromonas hydrophila*. The arrow indicates the antimicrobial protein; (**B**) Tryptic peptide map of the bacteriocin-like substance; (**C**) Identification of tryptic peptides derived from the bacteriocin-like substance by LC-MS/MS analysis. Peptides with ion scores greater than the identity threshold (score > 45) were regarded as identified peptides.

**Figure 2 molecules-24-02516-f002:**
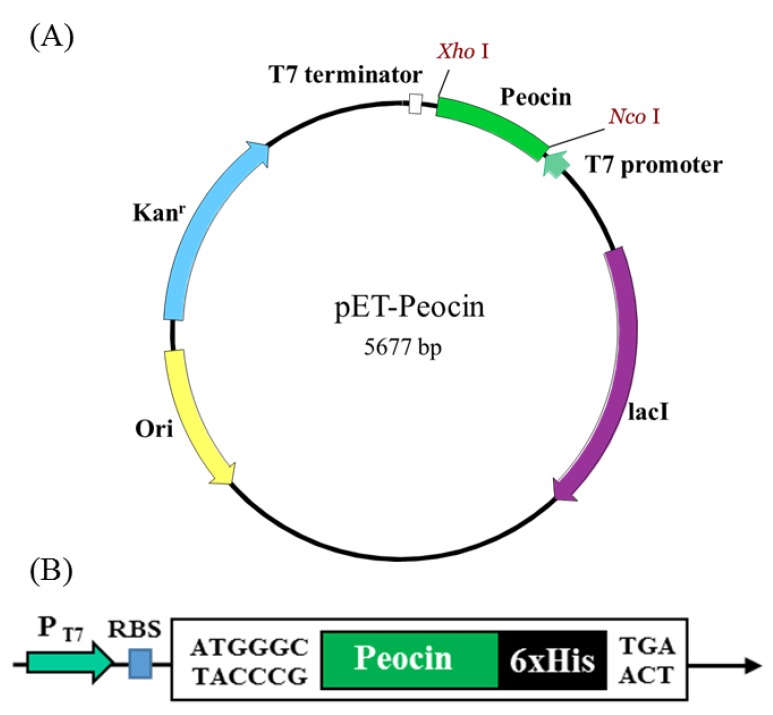
Construction of the recombinant expression plasmid pET-Peocin. (**A**) A synthetic DNA sequence encoding the peocin protein was cloned into pET28b with the *Nco*I and *Xho*I restriction sites. Kanr, kanamycin resistance; *lac*I, lac repressor; ori, origin of replication; (**B**) Diagram illustrating the main features of pET-Peocin. Expression of the peocin gene is controlled by the T7 promoter (P_T7_). RBS, ribosome-binding site.

**Figure 3 molecules-24-02516-f003:**
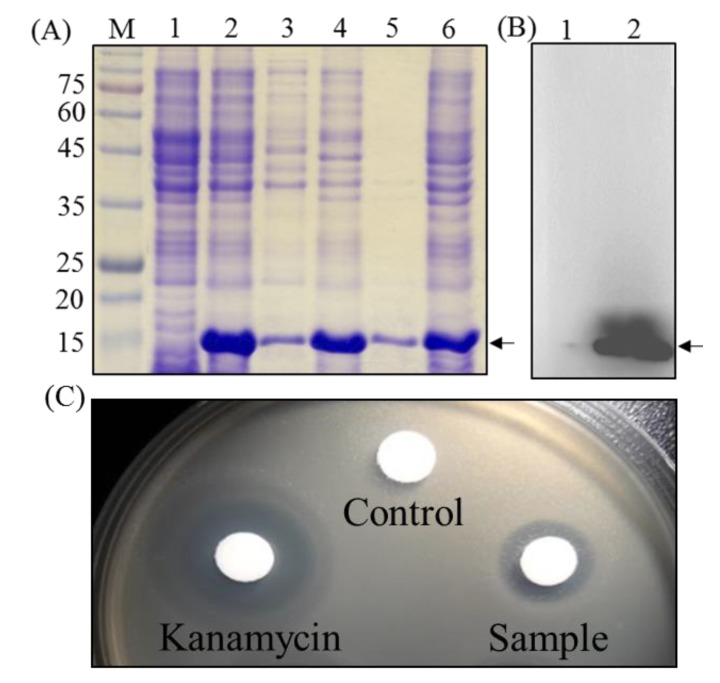
Expression and antimicrobial assay of recombinant peocin in *E. coli* BL21(DE3). (**A**) SDS-PAGE analysis of protein from the *E. coli* BL21(DE3)/pET-peocin host; (**B**) The recombinant peocin was identified by Western blotting using a 6x-His epitope tag antibody. Lane M: protein marker; 1: total protein without IPTG induction; 2: total protein with IPTG induction at 37 °C; 3 and 5: insoluble proteins with IPTG induction at 37 °C and 27 °C, respectively; 4 and 6: soluble proteins with IPTG induction at 37 °C and 27 °C, respectively; (**C**) Antimicrobial activity assay of recombinant peocin from the lytic supernatant of *E. coli* BL21(DE3)/pET-peocin. Control and sample indicate the 20 µL of lytic supernatant from *E. coli* BL21(DE3)/pET-peocin without and with IPTG induction, respectively. A 10 µg of kanamycin was used as positive control.

**Figure 4 molecules-24-02516-f004:**
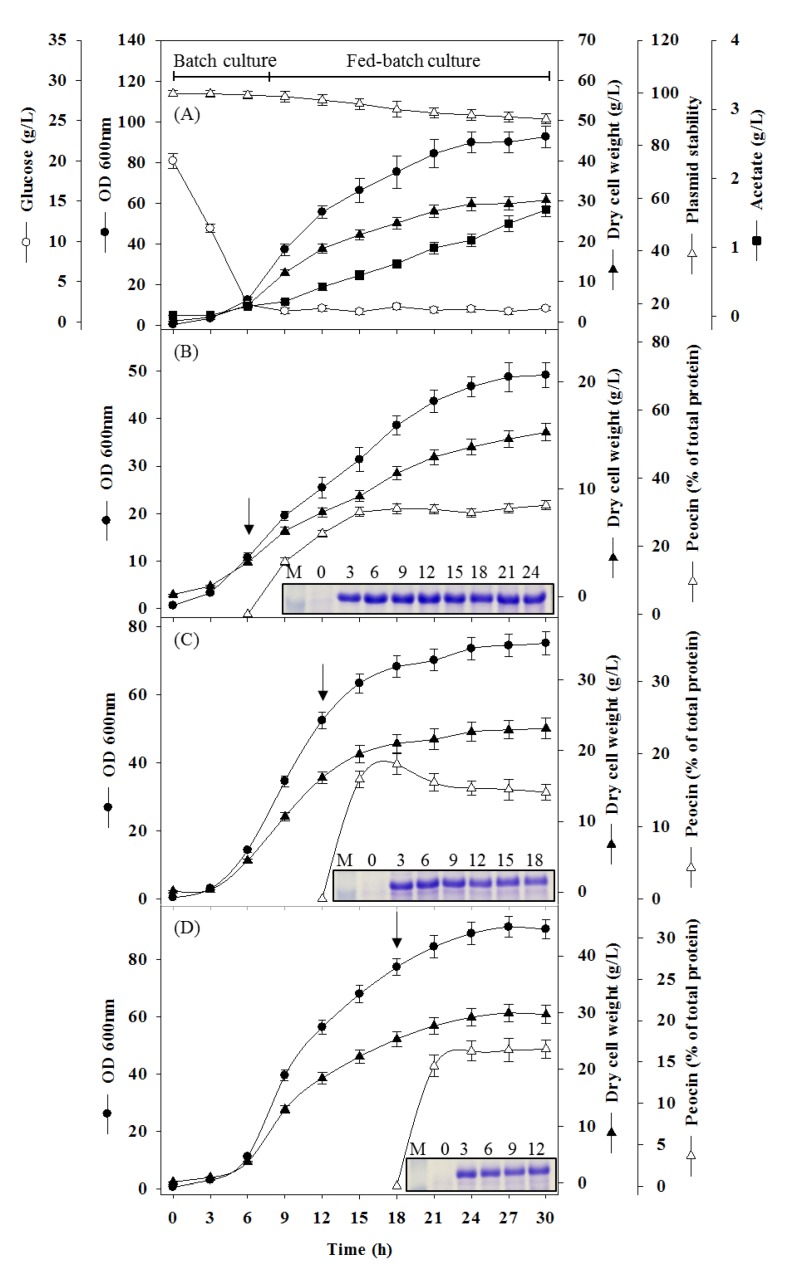
(**A**) Fed-batch cultivation of recombinant *E. coli* BL(DE3)/pET-peocin with a pH-stat without IPTG induction. Cell density OD_600_ (●), dry cell weight (▲), glucose (○), plasmid stability (△), and acetic acid (■). Fed-batch cultivation with IPTG induction at (**B**) early log phase (6 h), (**C**) midlog phase (12 h), and (**D**) late log phase (18 h). Cell density OD_600_ (●), dry cell weight (▲), and ggGH content (△). The arrow indicates the time of IPTG induction. SDS-PAGE for peocin content analysis is shown in the bottom right corner. Lane M indicates the molecular mass standard; the numbers indicate the hours post induction.

**Figure 5 molecules-24-02516-f005:**
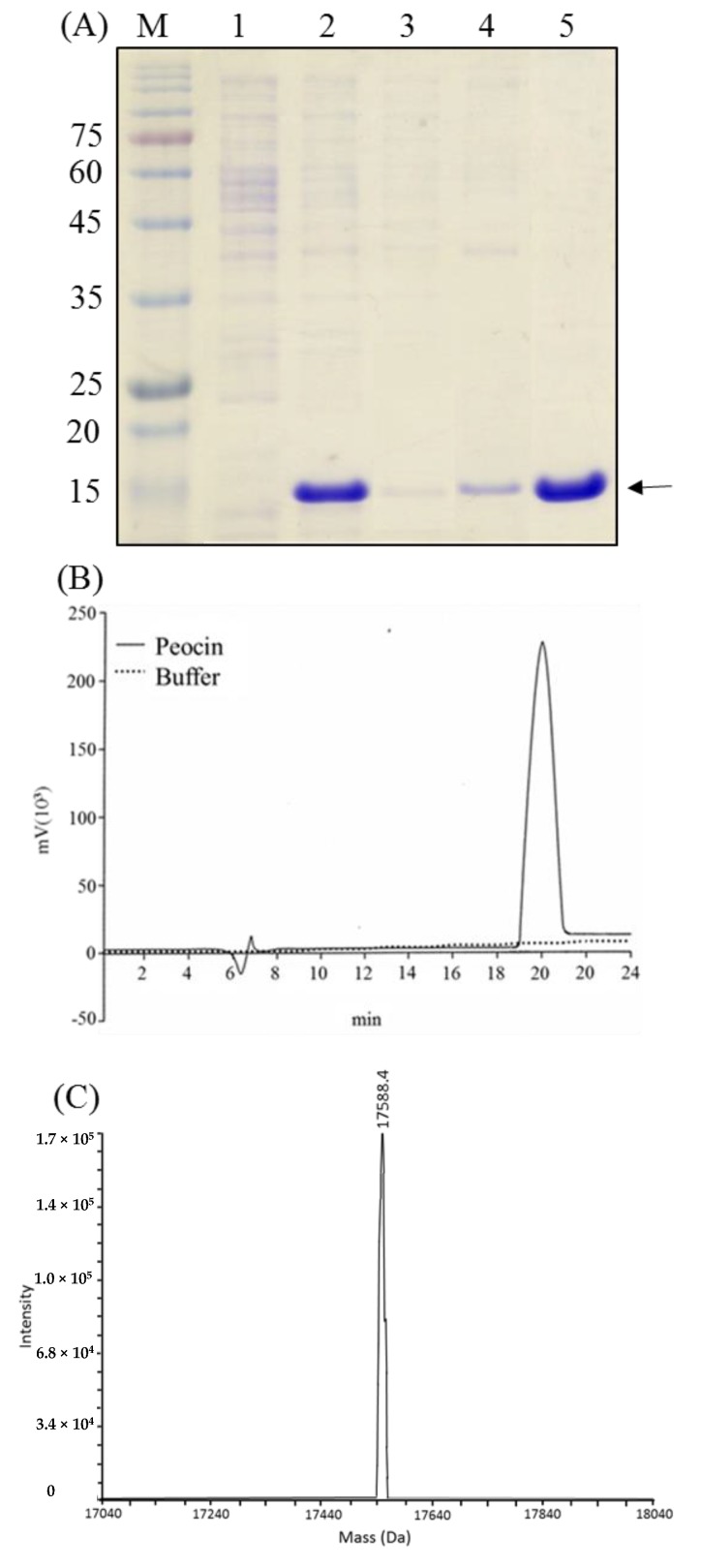
(**A**) Samples from each purification step of Ni-NTA affinity chromatography analyzed by 15% SDS-PAGE. Lane M: molecular mass standard; lanes 1 and 2, supernatant from *E. coli* BL21(DE3)/pET-peocin without and with IPTG induction, respectively; lane 3, flow through from the nickel affinity column; lane 4, flow through after washing with PBS containing 30 mM imidazole; lane 5, sample after eluting with PBS containing 250 mM imidazole and dialysis; (**B**) Reverse phase HPLC analysis of purified peocin; (**C**) Measurement of the molecular weight of purified peocin by MS analysis.

**Figure 6 molecules-24-02516-f006:**
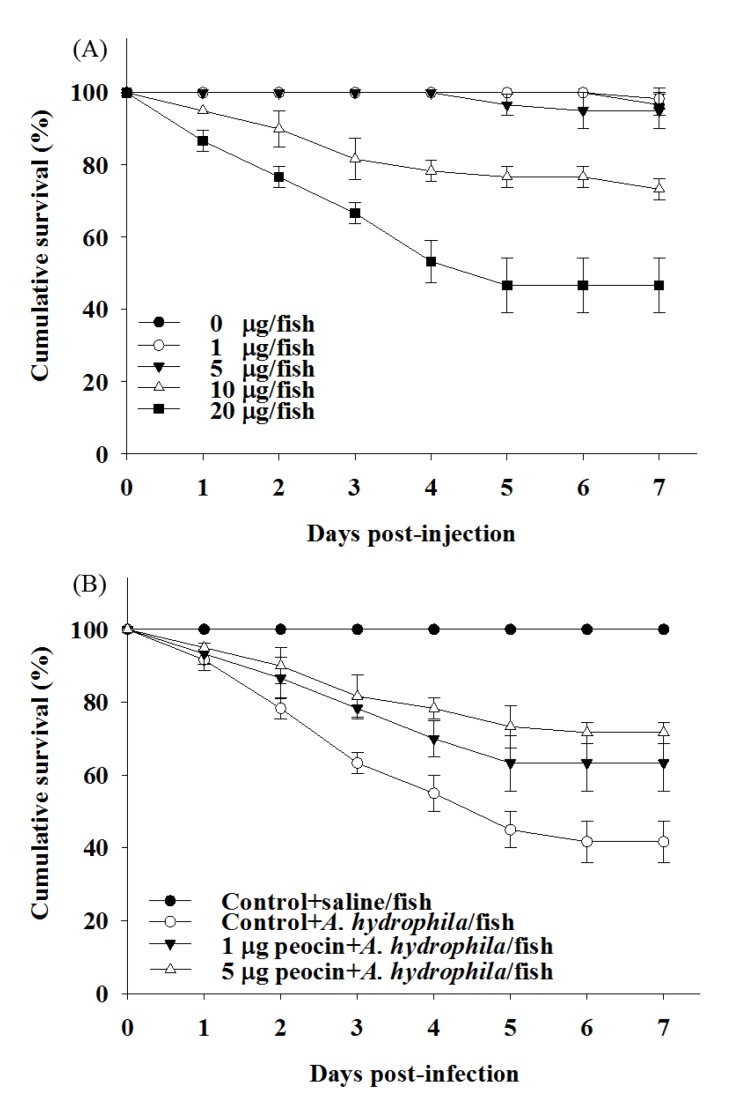
Cumulative survival of zebrafish injected with (**A**) various concentrations of purified peocin and (**B**) *A. hydrophila*. The daily survival rate of each group is recorded and is presented as the mean ± standard deviation (SD) from triplicate experiments.

**Table 1 molecules-24-02516-t001:** Protein recovery in a representative experiment.

Purification Step	Volume (mL)	Protein Conc. (mg/mL)	Amount of Total Protein (mg)	Amount of Peocin (mg)	Peocin Recovery (%)	Purity (%)
Supernatant of cell lysate ^a^	20	6.82	136.4	91.8	100	67.3
Ni-NTA affinity chromatography	15	4.28	64.3	59.4	64.8	92.3 ^b^
Dialysis	24	2.05	49.2	45.3	49.3	92.1 ^c^

^a^ 20-mL fed-batch culture was induced at the early late exponential phase, corresponding to 2.5 g of wet cell weight. ^b^ Determined by densitometric scanning of SDS-PAGE gel. ^c^ Determined by HPLC.

**Table 2 molecules-24-02516-t002:** Antimicrobial activity of purified peocin and antibiotics against diverse pathogens.

Pathogen	Purified Peocin (µg/mL)		Kanamycin (µg/mL)		Pathogen	Purified Peocin (µg/mL)		Kanamycin (µg/mL)	
	MIC	MBC	MIC	MBC		MIC	MBC	MIC	MBC
*Aeromonas hydrophila*	20	40	0.5	2.0	*Staphylococcus aureus*	20	50	0.5	2
*Streptococcus agalactise*	30	50	2.0	4.0	*Salmonella typhimurium*	30	60	1	2.0
*Streptococcus iniae*	30	60	3.0	6.0	*Listeria monocytogenes*	40	60	1	3.0
*Vibrio vulnificus ^a^*	30	40	Non ^b^	Non ^b^	*Methicillin-resistant* *S. aureus (MRSA) ^a^*	Non ^b^	Non ^b^	Non ^b^	Non ^b^
*Vibrio parahaemolyticus ^a^*	30	40	Non ^b^	Non ^b^	*Pseudomonas aeruginosa*	40	60	2.0	4.0
*Vibrio alginolyticus*	30	50	3.0	6.0	*Porphyromonas gingivalis*	30	60	2.0	3.0
*Debaryomyces hansenii*	50	80	Non ^b^	Non ^b^	*Prevotella nigrescens*	20	50	1	3.0
*Escherichia coli*	20	30	1.0	2.0	*Propionbacterium acnes*	20	40	1	2.0

^a^ Antibiotic-resistant pathogens. ^b^ Non indicates an antibiotic with no antimicrobial potency against the pathogen.

**Table 3 molecules-24-02516-t003:** Bacteriocin heterologous produced by *E. coli.*

Bacteriocin	Native Host	*E. coli* Strain	Vector	Culture Type	Bacteriocin Yield	References
Divercin 41 Class(IIa)	*C. divergens* V41	Origami (DE3)	pET32b	Fed-batch	74 mg/L	[33]
Enterocin P (Class IIa)	*Enterococcus faecium* P13	Tuner (DE3)	pETBlue-1	Shake-flask	0.006 mg/L	[34]
Pediocin PA-1 (Class IIa)	*P. acidilactici* PAC1.0	Origami (DE3)	pET-32b	Shake-flask	30 mg/L	[35]
Piscicolin 126 (Class IIa)	*C. piscicola* JG126	AD494(DE3)	pET-32b	Fed-batch	26 mg/L	[36]
Plantaricin E (Class IIb)	*L. plantarum*	BL21 (DE3)	pET32a	Shake-flask	15 mg/L	[37]
BtCspB	*B. thuringiensis* BRC-ZYR2	BL21 (DE3)	pET32a	Shake-flask	20 mg/L	[18]
Enterolysin A (Class III)	*E. faecalis*	SG13009[pREP4]	pQE-30 UA	Shake-flask	1.69 mg/g of wet cells	[38]
Peocin (Class III)	*P. ehimensis*	BL21 (DE3)	pET28b	Fed-batch	2260 mg/L	This work

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
