# Peer review of "Molecular Characterization and Heterologous Production of the Bacteriocin Peocin, a DNA Starvation/Stationary Phase Protection Protein, from Paenibacillus ehimensis NPUST1"

_molecules, 2019, doi:10.3390/molecules24132516_

Round 1
Reviewer 1 Report
The manuscript describes the identification of Peocin as a DNA_binding protein from starved cells (Dps). The manuscript also describes the production of recombinant Peocin and its antimicrobial activity. The is relatively straightforward biochemistry that is very well written. The authors add novel information to the field. I can not identify any significant weaknesses as written.
Author Response
Thank you for reviewing the manuscript and recognizing our study.
Reviewer 2 Report
The manuscript entitled “Molecular Characterization and Heterologous Production of the Bacteriocin Peocin, a DNA Starvation/Stationary Phase Protection Protein, from Paenibacillus ehimensis NPUST1” by C.-C. Tseng et al. continues a set of works devoted to structure-functional analysis of the novel antimicrobial protein from the bacteriocin family, in particular, in aspect to selection of fermentation conditions to significant increase a total protein yield in E. coli heterologous system, and estimation of its antimicrobial activity towards a set of aquatic, food spoilage, clinical and antibiotic-resistant microbes in vitro and in vivo assays to possible claiming this protein as potential substance for clinical and veterinary medicine. This paper is relatively well-structured and written, but som major points are available and to be corrected for further recommendation of this manuscript for publication in Molecules. A list of these remarks and advises is presented below.
1. Page 4, figure 1 – the authors are invited to explain which minor bands are visualized on the figure 1A in the range of 30-60 kDa? Compared with the target protein, their stain intensity is significant, that’s is why this is a question exists about total purity of the native BLIS isolated from the bacteria? Concerning figure 1C – where is a HPLC and mass spectra profile of the fragments obtained as result of tryptic cleavage? These data strongly needed to be added to the paper, for instance, in Supplementary.
2. Page 4, line 141 – the authors claim than more than 82.6% of the total recombinant protein was expressed in soluble form. A logical question assigned – what about the last 17.6%? Usually all recombinant expressed polypeptides are synthesized or in completely soluble, or in inclusion bodies. Please comment this moment in detailed.
3. Page 6, figure 3C – on the picture we can see antimicrobial activity of the protein studied is determined by inhibition zone applied by disc-diffusion assay. Where is positive control presented?
4. Page 8, figure 5 – there is only SDS-PAGE analysis of the purified protein demonstrated. But the required method for control of all recombinant proteins/peptides is HPLC, in particular, reversed-phase. This method is absolutely absent in this work. I expect, it has been to be involved. The graphical quality of figure 5B is very poor. Moreover it should be added a baseline for wide range of m/z (1-20 kDa). I suppose there are minor peaks are appeared which can be represented some admixtures and/or products of protein degradation.
5. Page 12 – concerning in vivo experiments…have the authors conducted a preliminary analysis of zebrafish in infection prior to assays? I understand that artificial infection has been made only A. hydrofilia culture, but it’s interested, what kind of contribution could be imported by other pathogenic microbiota in the final result?
6. Figure 6 – obliviously, peocin is toxic for zebrafish, maybe like another animals either. Moreover, compared with kanamycin, its effectiveness is very low: we can see more than 10-20 times more protein substance to achieve MIC and MBC. Both of these conclusions become to doubt that peocin will be attractive for further applied investigation, in particular, in aquaculture.
7. Discussion is written poorly. We have got a few literature cited. I suggest to add a table which is summarized all data about total yields of other bacteriocins. And because the peocin toxity is one of the critical points of this manuscript, I also recommend to the authors to include the text some information about in vitro and in vivo toxity of some another bacteriocins, especially, structure-closed to peocin.
Author Response
Thank you for your comments. The comments are helpful for improving the quality of our manuscript. Our responses to your comments are provided below.
1. Authors’ response
As description in Material and Methods section 4.2., the bacteriocin-like substance from cell-free culture broth after 96 h of cultivation was partially purified by ammonium sulfate precipitation. Culture broth comprise a variety of proteins such as proteins derived from medium and secreted from bacteria. Ammonium sulfate precipitation method can only remove partial some non-target protein to increase the purity of target protein, but it is difficult to obtain a pure target protein by a simple ammonium sulfate precipitation. Hence, it is reasonable to visualize other non-target proteins on SDS-PAGE, and they showed a molecule weight range from 30~60 kDa.
As reviewer’s comment, the fragments were obtained from tryptic cleavage. In accordance with the Reviewer’s comment, the LC/MS spectra data have been supplemented in supplementary Figure 1 (in revised manuscript lines 699)
2. Authors’ response
In fact, recombinant proteins are synthesized in a dynamic state. Review have indicated the newly synthesized recombinant polypeptide expressed in the microenvironment of E. coli may differ from that of the original source in terms of pH, osmolarity, redox potential, cofactors, protein concentration and folding mechanisms. Inclusion body (IB) formation results from an unbalanced equilibrium between protein aggregation and solubilization (Rosano and Ceccarelli, 2014). For example, report mention that expression of human leukiemia inhibitory factory (hLIF) produced in E.coli BL21(DE3) at 37°C revealing 78% of the recombinant hLIF resided in inclusion bodys and 22% of hLIF expressed as soluble protein. On lowering the expression temperature to 18°C, the solubility of recombinant hLIF increased to 80.1% (Jung et al., 2013). Study reported that cyanophycin granule polypeptide (CGP) expressed in E.coli reveling that the insoluble CGP/soluble CGP (iCGP/sCGP) ratio increased to 0.45 at a high temperature of 37 °C whereas sCGP wasfound dominant with an iCGP/sCGP ratio of 0.13 at a low temperature of 17 °C (Tseng et al. 2017). Thus, recombinant polypeptides are not really synthesized in completely soluble or in inclusion bodies. In the present study, the recombinant peocin with induction at 37°C was more than 82.6% of the total recombinant peocin was expressed in soluble form with induction at 37°C, indicating around 17.6% was expressed as inclusion body.
References:
Jung, A.S., Koo, B.K., Chong, S.H. et al. (2013). Soluble expression of human leukemia inhibitory factor with protein disulfide isomerase in Escherichia coli and its simple purification. PLOS one. 8(12):e83781.
Rosano, G.L. and Ceccarelli, E.A. (2014). Recombinant protein expression in Escherichia coli: advances and challenges. Front Microbiol. 17(5):172.
Tseng, W.C., Fang, T.Y., Hsieh, Y.C., Chen, C.Y. Li, M.C. (2017). Solubility and thermal response of fractionated cyanophycin prepared with recombinant Escherichia coli. Journal of Biotechnology. 249:59-65.
3. Authors’ response
In accordance with the Reviewer’s comment, the kanamycin used as positive control have been supplemented in the revised manuscript Figure 3C (Page 5).
4. Authors’ response
We agree the reviewer’s comment, thus the method for purity analysis of purified peocin by HPLC has been performed and supplemented in revised manuscript (513~517). As result shown in below, the purity of purified peocin was 92.1% and the data has been supplemented in Table 1. As to the graphical quality of figure 5B, the experiment was entrusted and perform by Biotechnology Company. We get a PDF report from company, and the figure was captured from PDF report. Thus, the graphical quality of figure 5B is poor. Although we contact with company and try to acquire the original graph, however the company didn’t keep the data due to the shelf life over three months. We apology for unable providing the original data of figure 5B.
Reverse phase HPLC analysis of purified peocin.
5. Authors’ response
Thank you for your comment. In the challenged test, a typical syndromes such as abnormal swimming behavior, swollen abdomen, hemorrhages in anus and abdomen can be observed in the A. hydrophila-infected zebrafish (shown in the photo below). Our previous study have demonstrated that the bacteria isolated from A. hydrophila-infected zebrafish almost are A. hydrophila. Although a few other bacteria can be seem on agar plate, but the amount almost can be ignored compare to the amount of A, hydrohplia (Faikoh et al., 2014). Reference
Faikoh, E.N., Hong, Y.H., Hu, S.Y. (2014). Liposome-encapsulated cinnamaldehyde enhances zebrafish (Danio rerio) immunity and survival when challenged with Vibrio vulnificus and Streptococcus agalactiae. Fish and Shellfish Immunology. 38:15-24.
Arrows indicated typical syndromes such as swollen abdomen and hemorrhages in anus and abdomen occurred in A. hydrophila-infected zebrafish
6. Authors’ response
We agree reviewer’s comment. We used an artificial injected-infection to perform the in vivo functional assay of recombinant peocin. The injected dose of peocin higher than 5 mg/fish is toxic to fish. However, the injection approach of peocin will not practical application in aquaculture. We assume that the crude peocin protein from E. coli can be used as feed additives or purified peocin can be used in processed aquatic food for preventing aquatic pathogen infection. As we known that miss use of antibiotics has led to the emergence of antibiotic-resistant pathogens, residual antibiotics in aquatic products, and alteration of the microbiota in aquaculture environments. In the present study, the point is not to compare antimicrobial activity of peocin with antibiotic kanamycin. We want to emphasis that peocin exhibit antimicrobial against diverse pathogens, in particular, antibiotic-resistant pathogens, and thus have potential alternative to antibiotics used in animal rearing and external biomedical product.
7. Authors’ response
In accordance with the Reviewer’s comment, the Table 3 summarized all data about total yields of other bacteriocin has been supplemented in Discussion section of revised manuscript (lines 332~347, and lines 412). The description of peocin toxicity has been supplemented in revised manuscript (line 358~365)
Reviewer 3 Report
Submitted manuscript reported the molecular characterization and heterologous production of the bacteriocin peocin from Paenibacillus ehimensis NPUST. Peocin was also tested as antimicrobial agent against diverse pathogens.
The manuscript was clear and the English provided was very professional. Details about biological assays were correctly given. In my view, the manuscript should be accepted as it is.
Author Response

(The authors gave the same response as above.)

Round 2
Reviewer 2 Report
Thank to the authors for the comments provided. I'm almost sutisfied all of tham, but I've to insist on my remark concerning HPLC profile of peocin isolation. I'm completely sure that only table data are not enough for these culculated conclutions presented in the Table 2. Please add the HPLC profile to the text or as Supplementary Material.
Author Response
Thank you for your comments. The comments are helpful for improving the quality of our manuscript.
We agree the reviewer’s comment, thus the HPLC data has been supplemented in Fig 5B (in revised manuscript lines 213 and line 222).